**DOI: 10.1038/ncomms14486**　　**OPEN**

# Tailoring the thermal and electrical transport properties of graphene films by grain size engineering

Teng Ma[1], Zhibo Liu[1], Jinxiu Wen[2], Yang Gao[1], Xibiao Ren[3], Huanjun Chen[2], Chuanhong Jin[3], Xiu-Liang Ma[1], Ningsheng Xu[2], Hui-Ming Cheng[1] & Wencai Ren[1]

Understanding the influence of grain boundaries (GBs) on the electrical and thermal transport properties of graphene films is essentially important for electronic, optoelectronic and thermoelectric applications. Here we report a segregation–adsorption chemical vapour deposition method to grow well-stitched high-quality monolayer graphene films with a tunable uniform grain size from $\sim 200\,nm$ to $\sim 1\,\mu m$, by using a Pt substrate with medium carbon solubility, which enables the determination of the scaling laws of thermal and electrical conductivities as a function of grain size. We found that the thermal conductivity of graphene films dramatically decreases with decreasing grain size by a small thermal boundary conductance of $\sim 3.8 \times 10^9\,W\,m^{-2}\,K^{-1}$, while the electrical conductivity slowly decreases with an extraordinarily small GB transport gap of $\sim 0.01\,eV$ and resistivity of $\sim 0.3\,k\Omega\,\mu m$. Moreover, the changes in both the thermal and electrical conductivities with grain size change are greater than those of typical semiconducting thermoelectric materials.

[1] Shenyang National Laboratory for Materials Science, Institute of Metal Research, Chinese Academy of Sciences, 72 Wenhua Road, Shenyang 110016, China. [2] State Key Laboratory of Optoelectronic Materials and Technologies, Guangdong Province Key Laboratory of Display Material and Technology, School of Physics and Engineering, Sun Yat-sen University, Guangzhou 510275, China. [3] State Key Laboratory of Silicon Materials, School of Materials Science and Engineering, Zhejiang University, Hangzhou 310027, China. Correspondence and requests for materials should be addressed to W.R. (email: wcren@imr.ac.cn) or to H.-M.C. (email: cheng@imr.ac.cn).

Graphene has attracted increasing interest because of the extraordinary properties of its defect-free pristine form, such as the highest known carrier mobility, record thermal conductivity and extremely high mechanical strength[1–3]. However, large-area graphene films produced by scalable methods, such as chemical vapour deposition (CVD), usually have various defects, especially grain boundaries (GBs)[4–13], forming a polycrystalline structure. Moreover, the GBs are formed randomly during CVD growth[5,6,8,12,13]. Therefore, in addition to studies of individual GBs, understanding the influence of grain size on the overall electrical and thermal transport properties of graphene films on a large scale is not only fundamental but also technologically important in order to tune their properties for electronic, optoelectronic and thermoelectric applications[12–24]. These studies strongly depend on the controlled synthesis of graphene films with tunable and uniform grain size that is smaller than the phonon and electron mean free paths (∼ a few hundreds of nanometres) because the contributions to electrical and thermal transports due to scattering from GBs are more significant in this range.

From the point of view of crystal growth, it is equally difficult to reduce and increase nucleation density on metals by CVD to fabricate large-size single-crystal graphene and polycrystalline graphene with nano-sized grains, respectively, while keeping monolayer growth of graphene. For surface adsorption growth on the commonly used Cu with a low carbon solubility, a high-concentration carbon source and/or defective substrates have usually been used to obtain a high domain density, however, these conditions led to the formation of multi-layer graphene domains[10,25,26]. It is well known that the graphene films segregated from Ni with a high carbon solubility are usually nonuniform multi-layers[27–29]. As a result, the polycrystalline graphene films prepared so far usually have a grain size ranging from ∼1 μm to ∼1 mm (refs 4–13), which is larger than the electron and phonon mean free paths (a few hundreds of nanometres)[21], and/or have very broad grain size distributions[5,6]. This strongly hinders the experimental studies on the real influence of grain size on the electrical and thermal transport properties of graphene films.

It has been theoretically predicted that electrical transport in graphene could be markedly altered by electron scattering at GBs[14,15]. Consistent with these predictions, many experimental studies on individual GB have shown that GBs can greatly impede electronic transport, thus degrading the carrier mobility and electrical conductivity of graphene[4,10–13], although a few experiments have shown that perfect inter-grain connectivity at GBs retains the remarkable electrical conductance of graphene[7,8]. However, the electrical measurements on graphene films have shown no strong correlation between the average grain size and the overall electron mobility[5,9]. The present studies on the influence of GBs on the thermal transport of graphene have been mainly limited to theoretical works, and different calculation methods have led to contradictory conclusions. Some theoretical calculations[20,22] have suggested that the thermal transport in polycrystalline graphene could be significantly degraded when the grain size is smaller than a few hundred nanometres, while others suggested that all types of GBs have excellent thermal transport[19]. Experimentally, the influences of the degree of disorders on the thermal and electrical conductivities have been investigated[30] and recent thermal transport measurements on individual GB have shown that a single GB can significantly decrease the thermal conductivity of graphene[31]. However, the influence of grain size on the overall thermal conductivity of graphene films remains unknown.

Here we have developed a segregation–adsorption CVD (SACVD) method to achieve a great increase in the nucleation density of graphene (by segregation) and monolayer growth (by surface adsorption) simultaneously, by using a Pt substrate with medium carbon solubility. As a result, we can easily grow well-stitched high-quality monolayer graphene films with a tunable uniform grain size from ∼200 nm to ∼1 μm, which have never been achieved before by the present conventional CVD methods based on either surface adsorption[4–13,27,32] or segregation mechanism[27–29]. Using these materials, we determined the scaling laws of thermal and electrical conductivities of graphene films as a function of grain size. It was found that the thermal conductivity of graphene films dramatically decreases with decreasing grain size by a small thermal boundary conductance of ∼$3.8 \times 10^9$ W m$^{-2}$ K$^{-1}$, while the electrical conductivity slowly decreases with an extraordinarily small GB transport gap of ∼0.01 eV and GB resistivity of ∼0.3 kΩ μm. Moreover, both the thermal and electrical conductivities of graphene change more significantly with grain size change than that of typical thermoelectric materials[33–35].

## Results

**SACVD growth process**. Figure 1a illustrates the fabrication process of polycrystalline graphene films by SACVD. First, we used a relatively high flow rate of methane mixed with hydrogen to rapidly grow a monolayer dominate graphene film on a Pt substrate by a surface growth mechanism (Fig. 1b, the first step). During this process, some carbon atoms were dissolved in the Pt substrate (Supplementary Figs 1 and 2, and Supplementary Note 1) because of the medium carbon solubility of Pt (0.07 wt.%) (ref. 36), which is higher than Cu (0.008 wt.%) but lower than Ni (0.3 wt.%) at 1,000 °C (ref. 27). Such medium carbon solubility allows that the growth behaviour of graphene can be tuned between surface adsorption and segregation. We then changed the atmosphere to pure argon to etch the graphene film formed on the surface into the bulk (Fig. 1c, the second step). After this, we induced the segregation of the dissolved carbon atoms by re-introducing a trace of hydrogen (Supplementary Figs 1–5 and Supplementary Notes 1 and 2), and a large number of small graphene domains appeared (Fig. 1d, the third step, Supplementary Note 2). Finally, we introduced a low flow rate of methane to induce surface growth of the graphene domains to form continuous monolayer polycrystalline films (Fig. 1e, Supplementary Fig. 6, the fourth step and Supplementary Note 3).

Interestingly, we can easily obtain a very high domain density that is suitable for growing monolayer graphene films with a grain size smaller than 1 μm by this SACVD method (Fig. 2a–d). The reaction temperature in the segregation process is the only factor that determines the domain density, and this is increased by decreasing the growth temperature (Fig. 2a–d and Supplementary Fig. 4). With reaction temperatures of 900, 950, 1,000 and 1,040 °C, monolayer graphene domains with respective densities of 96 ± 13, 18 ± 6, 11 ± 3 and 4 ± 2 μm$^{-2}$ were obtained (Fig. 2a–d). The corresponding mean domain sizes are ∼50 (Figs 1d and 2a), 100 (Fig. 2b), 200 (Fig. 2c) and 500 nm (Fig. 2d). Moreover, the domain density is entirely unrelated to the growth atmosphere, including the flow rates of hydrogen, argon and methane. In sharp contrast, such high-density monolayer graphene domains cannot be achieved by either surface adsorption growth on Cu[10,25–27] or segregation growth on Ni[27–29], as mentioned above. In our method, the use of Pt with medium carbon solubility allows the dissolution of a small amount of carbon, which is the key to obtaining a high-density monolayer of graphene domains by subsequent segregation.

**Structural characterization**. We used dark-field transmission electron microscopy (TEM)[5,6] to determine the grain size of the

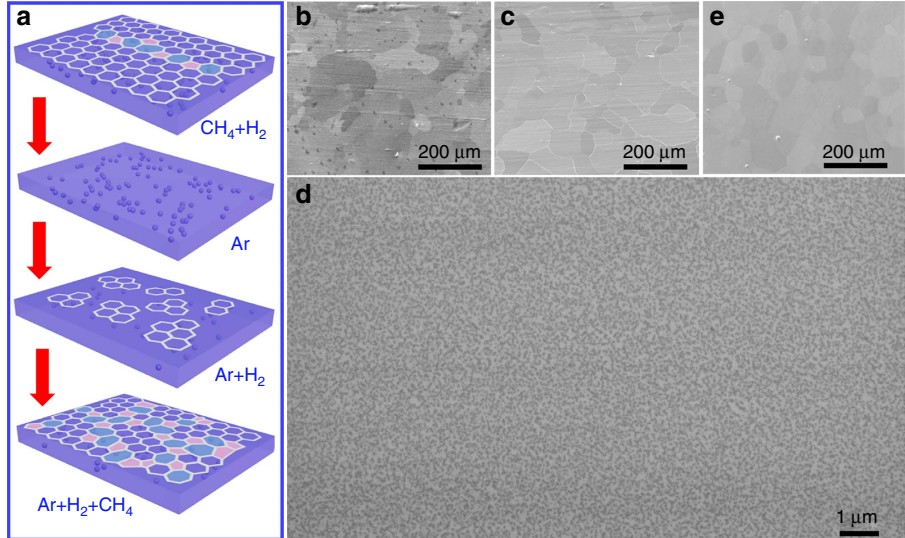

**Figure 1 | SACVD growth of polycrystalline graphene films with well-controlled grain sizes. (a)** Schematic for the fabrication process of a polycrystalline graphene film. **(b)** Scanning electron microscope (SEM) image of a graphene film, mostly monolayer, grown on Pt with a mixture of hydrogen (700 standard-state cubic centimetre per minute, sccm) and methane (7 sccm) for 10 min. **(c)** SEM image of the Pt substrate in **b** after treating with pure argon (700 sccm) for 20 min, showing that the graphene film has disappeared. **(d)** SEM image of the Pt substrate in **c** after treating with a trace of hydrogen (5 sccm) for 20 min, showing that many small graphene domains have appeared. **(e)** SEM image of a monolayer polycrystalline graphene film formed from **d** by introducing a low flow rate of methane (0.1 sccm) for 1 h. The reaction temperature was all 900 °C in above cases.

graphene films formed from isolated domains with different densities. To do this, the graphene films were first transferred onto a TEM grid with $2 \times 2\,\mu m^2$ circular holes covered with amorphous carbon. We then used different objective aperture filters to image the grains with different lattice orientations. Finally, the obtained multiple dark-field images were coloured with different colours and overlaid to form a complete map of the films, as shown in Fig. 2e–h. It can be clearly seen that the graphene films consist of high-quality grains with different orientations. TEM and Raman spectroscopy measurements show that all the grains are perfectly stitched together without any gaps (Supplementary Figs 7–11). We further performed aberration-corrected high-resolution TEM (HRTEM) measurements to obtain atomic-resolution structure information of the GBs. As shown in Fig. 2i,j and Supplementary Fig. 12, the GBs exhibit atomically sharp interface regions by chains of pentagons and heptagons embedded in the hexagonal lattice of graphene without overlapping, buckling and other defects. Note that a low flow rate ratio of methane to hydrogen was used during the surface adsorption growth process. The resulting slow growth rate facilitates the relaxation of metal-carbon system towards thermal equilibrium during growth, and consequently enables the perfect stitching of high-density graphene domains to form high-quality monolayer graphene films (Supplementary Figs 7 and 8).

We obtained histograms of grain sizes by measuring more than 100 grains for each sample (Fig. 2k–n). The mean grain sizes, defined as the square root of the grain area, are $1,013 \pm 90$, $721 \pm 79$, $470 \pm 74$ and $224 \pm 73$ nm. It is important to note that these sizes are much smaller than the typical grain size of the graphene films reported so far (usually larger than $1\,\mu m$) (refs 4–13), and smaller than or similar to the electron and phonon mean free paths[21]. Moreover, graphene films prepared under the same conditions show the same grain size distribution, that is, the process produces reproducible results. This highly reproducible synthesis of graphene films with a uniform mean grain size, smaller than the electron and phonon mean free paths, and perfect stitching of the GBs, opens up the possibility of investigating the real influence of grain size on the electrical and thermal transport in graphene.

**Thermal transport measurements.** Confocal micro-Raman spectroscopy is an efficient method for measuring the thermal conductivity of suspended graphene. Its value is extracted from the dependence of the Raman G or 2D peak frequency on the excitation laser power[37,38]. Here, we used the 2D peak shift to determine the graphene temperature because of its higher temperature sensitivity than the G peak[39]. Before thermal transport measurements, we first characterized the transferred graphene films on $SiO_2$/Si holey substrates (circular holes: $5\,\mu m$ in diameter, 290 nm in depth) to make sure that the suspended area is intact. The SEM image shows that most area of the substrate is covered by graphene without visible cracks (Fig. 3a). Figure 3c shows a $40 \times 40\,\mu m^2$ 2D peak intensity map of a graphene film with the corresponding optical image shown in Fig. 3b. It can be clearly seen that most of the suspended graphene films exhibits a uniform and much stronger 2D peak than the supporting area without the D peak (Fig. 3c and Supplementary Fig. 13), indicating that they are intact and have high quality. For thermal measurements, a 532 nm laser beam was focused on the centre of the suspended graphene film to obtain the power coefficient or on the supported graphene on $SiO_2$/Si to obtain the temperature coefficient, as reported by Balandin et al[38]. The thermal conductivity ($\kappa$) of the graphene films was calculated by $\kappa = \chi(1/2h\pi)(\delta w/\delta P)^{-1}$, where $\delta w$ is the shift of 2D peak position due to the change of heating power $\delta P$ on the sample, $\chi$ is the 2D peak temperature coefficient, and $h$ is the thickness of the graphene film.

Figure 3d shows the Raman spectra of the graphene films with $\sim 200$ nm-size grains excited by lasers with different powers. It is interesting to see that the D peak intensity and $I_D/I_G$ increase sharply while the G peak intensity decreases dramatically when the laser power is larger than 1.2 mW (Fig. 3d and Supplementary Fig. 14). The 2D peak upshifts and dramatically increases in intensity with the laser power until 1.2 mW (Fig. 3e,f). However, when further increasing the laser power, the 2D peak intensity decreases and the corresponding peak position changes randomly. Moreover, the intensities of 2D and G peak and $I_D/I_G$ cannot recover their original values

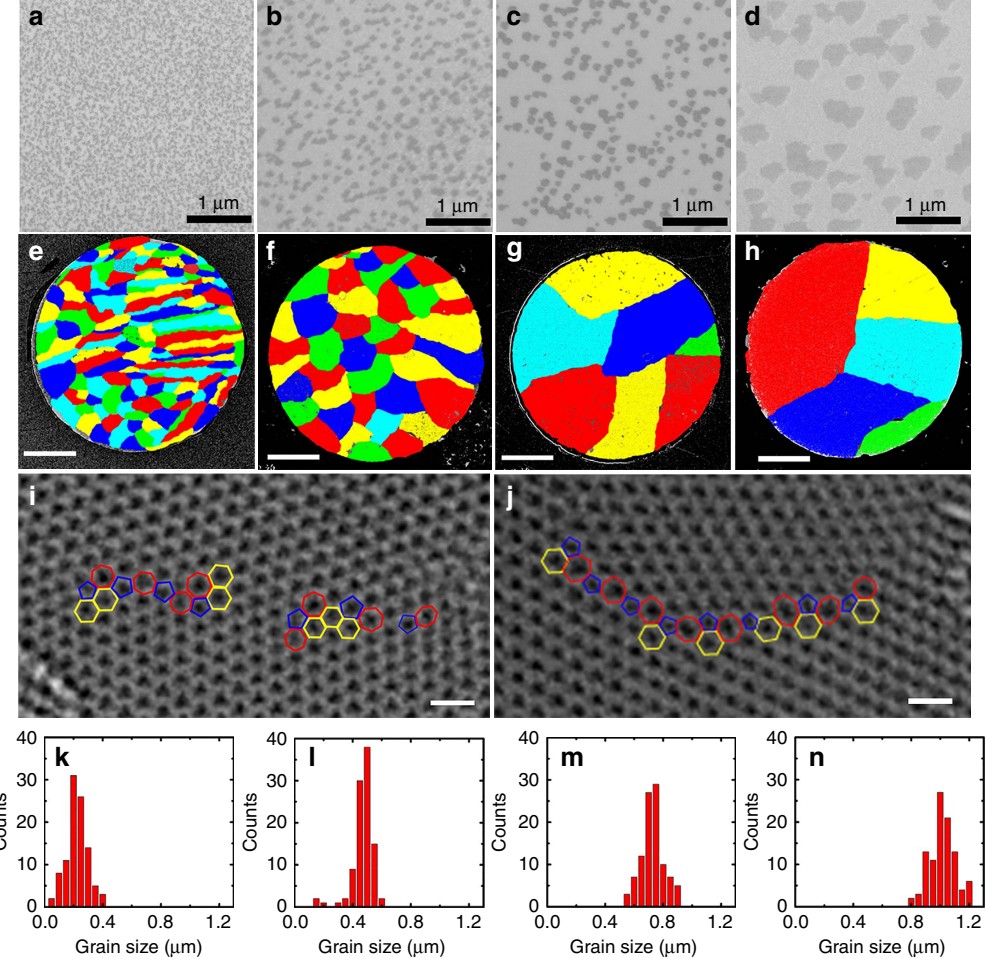

**Figure 2 | Structural characterization of graphene domains and films.** (**a–d**) SEM images of graphene domains obtained with a segregation temperature of 900, 950, 1,000 and 1,040 °C, showing that the domain density decreases with segregation temperature. (**e–h**) False-colour, dark-field image overlays of the graphene films formed by growth and stitching of the graphene domains in **a–d**. Scale bars, 500 nm. (**i,j**) High-magnification HRTEM images of graphene films with grain size of ~200 and ~700 nm, respectively. The pentagons (blue), heptagons (red) and hexagons (yellow) in the GBs are outlined. All images were processed with an improved Wiener-filtering to remove the noises. Scale bars, 1 nm. (**k–n**) Histograms of grain sizes of the graphene films in **e–h**, showing that the grain size is very uniform for each sample.

at the same laser power when the laser power was decreased (Fig. 3e and Supplementary Fig. 14). These phenomena indicate that the suspended graphene films with small grains have been destroyed by the high-power laser. According to the 2D peak shift (13.8 cm$^{-1}$) and the extracted temperature coefficient (0.039 cm$^{-1}$ K$^{-1}$), we estimated that the temperature at the GBs of the graphene film with ~200 nm grains reached 650 K when the laser power was 1.2 mW. Combined with the Raman spectra evolution, the physical origins of the D and G peak[40] and the high activity of GB[12,41], we suggest that this temperature jump results in the breaking of the graphene film at GB due to strong thermal vibration[42]. In sharp contrast, the suspended graphene films with ~1 mm grains (no GBs across the suspended area) remained intact with a low D peak even when illuminated by a laser of 2.8 mW for 10 s (Supplementary Fig. 15). The above results give direct evidence that GBs greatly reduce the thermal conductivity of graphene.

Figure 4a shows the thermal conductivity of the polycrystalline graphene films ($\kappa$) as a function of grain size ($l_g$). It is clear that the thermal conductivity increases exponentially from ~610 to ~5,230 W m$^{-1}$ K$^{-1}$ when the grain size is increased

from ~200 nm to ~10 μm. In fact, the graphene films with grain size larger than ~5 μm (the size of the suspended area) all show a similar thermal conductivity of ~5,200 W m$^{-1}$ K$^{-1}$ (thermal conductivity within the grain, $\kappa_g$), which is similar to the value reported for pristine graphene made by mechanical exfoliation[38]. This confirms that our measurement method is appropriate and our SACVD grown samples have very high quality, which rules out the influence of defects on the thermal conductivity and ensures that the thermal conductivity change is intrinsically related to GBs. On the basis of the kinetic theory of phonon transport[21], the effective phonon mean free path is given by $l_{eff}^{-1} = l_{ph-ph}^{-1} + l_g^{-1}$, where $l_{ph-ph}$ denotes the phonon–phonon scattering length and $l_g$ is the scattering length due to the boundaries (that is, grain size)[18]. Consistent with this, it is very interesting to note that the inverse of thermal conductivity ($\kappa^{-1}$) versus the inverse of grain size ($l_g^{-1}$) can be well fit by $\kappa^{-1} = \kappa_g^{-1} + (l_g G)^{-1}$, where $\kappa_g$ is the thermal conductivity within the grain (~5,200 W m$^{-1}$ K$^{-1}$) and $G$ is the boundary conductance[18]. The extracted thermal boundary conductance is ~3.8 × 10$^9$ W m$^{-2}$ K$^{-1}$, which is consistent with the theoretical value obtained using non-equilibrium Green's functions (3–8 × 10$^9$ W m$^{-2}$ K$^{-1}$) (ref. 20). The scaling law can be written

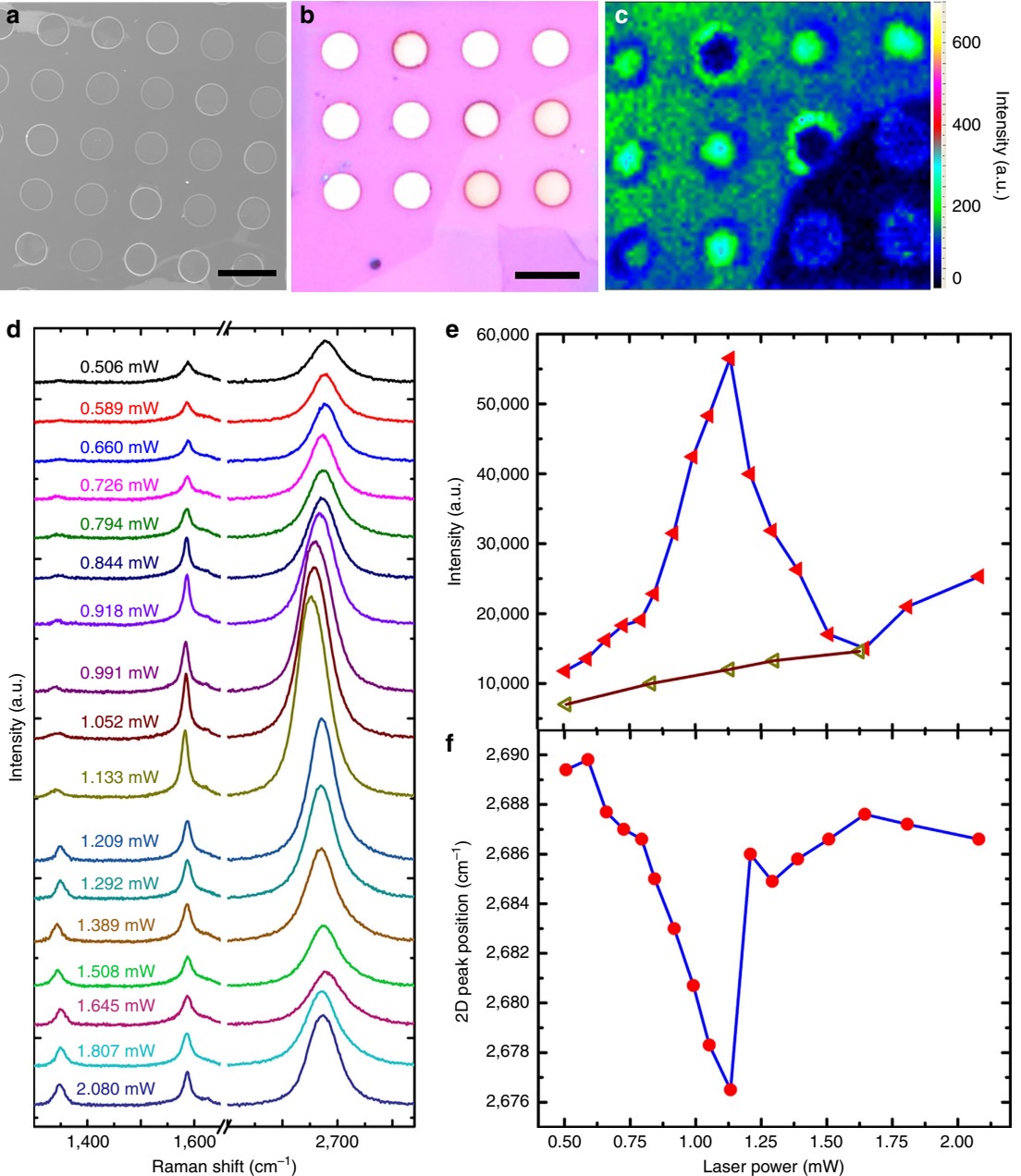

**Figure 3 | Thermal transport of graphene films with ∼200 nm-sized grains. (a)** SEM image of a polycrystalline graphene film on a holey SiO$_2$/Si substrate. Scale bar, 10 μm. **(b)** Optical image of a polycrystalline graphene film transferred onto a holey SiO$_2$/Si substrate. Scale bar, 10 μm. **(c)** Raman map of the polycrystalline graphene film shown in **b**, and the typical Raman spectra are shown in Supplementary Fig. 12. **(d)** Raman spectra of the polycrystalline graphene film excited with different power lasers. **(e,f)** Intensity **(e)** and position **(f)** of the 2D peak as a function of laser power.

as $\kappa^{-1} = 0.26\, l_{\mathrm{g}}^{-1} + 0.19$. As we know, the scattering of phonons within the grains primarily determine the thermal conductivity of the polycrystalline graphene when the grains are large in size, while the contribution to thermal conductivity due to scattering from GBs becomes more significant with decreasing grain size[18]. Using the above scaling law, we estimated that the critical size of grains below which the contribution from the GBs becomes comparable to the scattering from the grain is $l_{\mathrm{g}} = \kappa_{\mathrm{g}}/G \approx 1.4\,\mu\mathrm{m}$.

**Electrical transport measurements.** To evaluate the influence of GBs on electrical properties, we used a four-probe station to measure the sheet resistances of the graphene films with different grain sizes (Fig. 4c), and dozens of positions were measured for each sample (2 cm × 2 cm). We fit the data using modified Arrhenius equation[43] $\sigma = \sigma_0\, \exp\{- E_{\mathrm{a}}/[RT(l_{\mathrm{g}} + c)]\}$ (Fig. 4d), where $\sigma$ is the electrical conductivity of the polycrystalline graphene films, $\sigma_0$ is the electrical conductivity within the grain, $E_{\mathrm{a}}$ is the GB transport gap (the energy that is needed to overcome for the charge carrier transmitting through the GB region), $R$ is the universal gas constant, $T$ is the absolute temperature, $l_{\mathrm{g}}$ is the grain size and $c$ is the correction value. The fitting gives $\sigma_0 \approx 2.85 \times 10^6\,\mathrm{S\,m^{-1}}$ and $E_{\mathrm{a}} \approx 0.01\,\mathrm{eV}$. Note that the GB transport gap extracted here is dramatically smaller than the theoretically predicted value for asymmetric GBs (0.3–1.4 eV)

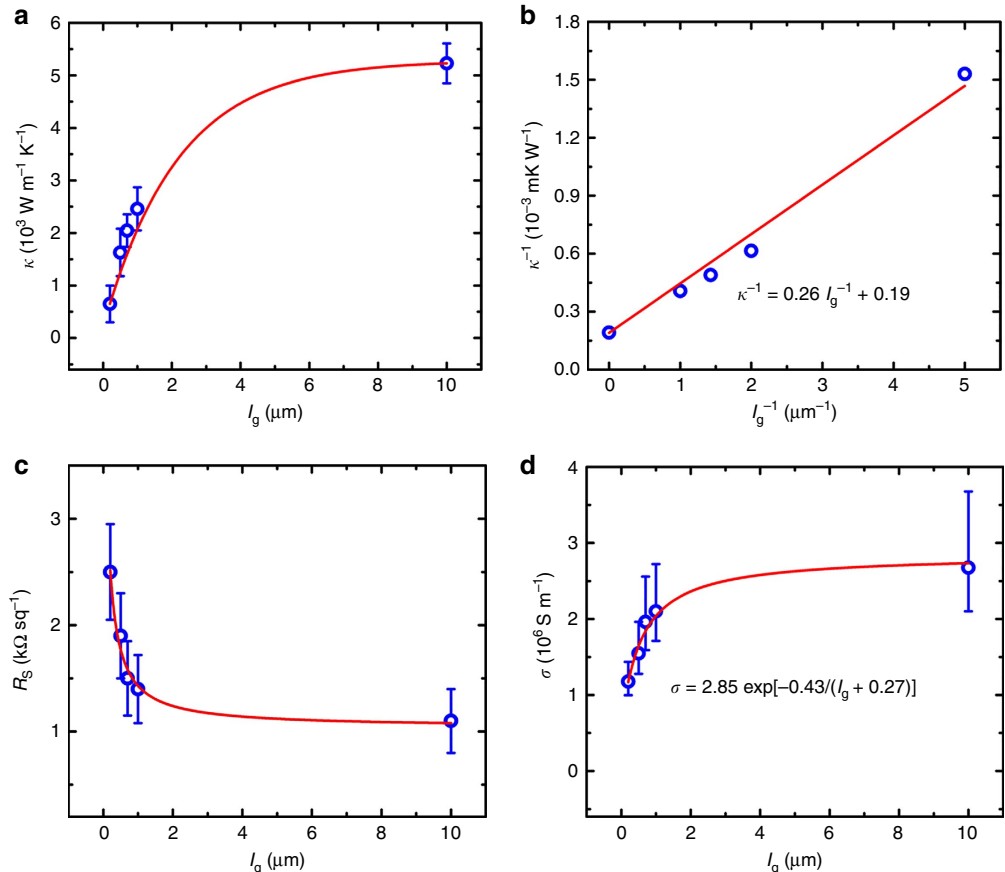

**Figure 4 | Thermal and electrical transport of the graphene films with different grain sizes.** (**a**) Thermal conductivity as a function of grain size with a fit (red curve). The error bars (standard error of the mean, s.e.m.) represent the thermal conductivity variation measured for the same sample. (**b**) The inverse of thermal conductivity as a function of the inverse of grain size with a fit (red curve), showing a linear relationship. (**c**) Sheet resistance as a function of grain size with a fit (red curve). (**d**) Electrical conductivity as a function of grain size with a fit (red curve), showing an exponential relationship. The error bars (s.e.m.) in **c** and **d** represent the electrical conductivity variation measured for the same sample and the samples prepared with the same conditions.

(ref. 14). Using this scaling law, we found that the GBs begin to dominant the electrical conductivity of the poly-crystalline graphene films only when the grain size is smaller than $l_g \approx 0.8 \,\mu m$. We also fit the data using the equation $R_s = R_s^G + \rho_{GB}/l_g$ (Supplementary Fig. 16)[13], where $R_s$ is the sheet resistance of the polycrystalline graphene films, $R_s^G$ is the sheet resistance within the grain, $\rho_{GB}$ is the GB resistivity and $l_g$ is the grain size. The fitting gives $R_s^G \approx 0.98 \,k\Omega \,sq^{-1}$ and $\rho_{GB} \approx 0.33 \,k\Omega \,\mu m$. It is worth noting that the GB resistivity extracted here is smaller than those reported previously, typically larger than $0.5 \,k\Omega \,\mu m^{4,8,13,44}$, further confirming the perfect stitching of neighbouring grains in our graphene films. Both the small GB transport gap and GB resistivity suggest the weak influence of grain size on the electrical conductivity, which is in sharp contrast to thermal conductivity. As shown in Supplementary Fig. 17, when the mean grain size is increased from $\sim 200 \,nm$ to $\sim 1 \,mm$ (five orders of magnitude increase), there is only a fourfold increase in electrical conductivity. The above results suggest that increasing grain size is not an efficient way to improve the electrical conductivity of graphene for transparent conductive electrode applications when the grain size is larger than $1 \,\mu m$.

## Discussion
To further compare the influence of GBs on the thermal/electrical conductivity of graphene films, we plotted (Fig. 5) the thermal/electrical conductivity change rate as a function of grain

size change rate ($\Delta l_g \, l_g^{-1}$). Note that the thermal conductivity change rate ($\Delta \kappa \, \kappa^{-1}$) increases linearly with grain size change rate (Fig. 5a), while electrical conductivity change rate ($\Delta \sigma \, \sigma^{-1}$) increases exponentially with grain size change rate (Fig. 5b). More importantly, the thermal conductivity change rate of graphene is dramatically larger than the electrical conductivity change rate (Fig. 5). According to the scaling law of thermal conductivity as a function of grain size shown above, the thermal conductivity of graphene films with a grain size of 5 nm is extrapolated to be $\sim 19.2 \,W \,m^{-1} \,K^{-1}$, a $\sim 300$ times decrease compared with pristine graphene. However, the electrical conductivity is extrapolated to be $\sim 5.9 \times 10^5 \,S \,m^{-1}$ based on the modified Arrhenius equation with better fitting than the equation $R_s = R_s^G + \rho_{GB}/l_g$, only a $\sim 10$ times decrease compared with graphene with a millimetre grain size. Therefore, nano-crystallization should be an efficient way to tune the electrical and thermal conductivities of polycrystalline graphene films for thermoelectric applications if graphene could be used in thermoelectric materials in the future as predicted[45,46]. Even for the graphene films with a 1-nm grain size, both the thermal and electrical conductivities are much larger than those of amorphous carbon although its grain size is much smaller[17], indicating that the disorder within grain plane may have much stronger influence on the electrical and thermal properties of carbon materials.

We also compared the thermal/electrical conductivity change rate of graphene with those of some typical metals (Au, Ag, Cu and Al)[47–49] and semiconducting thermoelectric materials

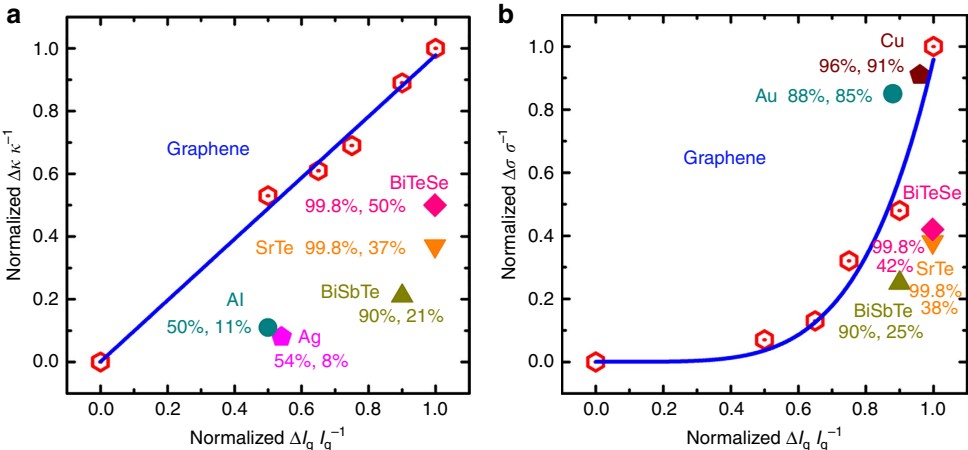

**Figure 5 | Thermal/electrical conductivity change rate of graphene with grain size change rate.** (**a**) Thermal conductivity change rate of graphene as a function of grain size change rate with a fit (blue curve), showing a linear relationship. (**b**) Electrical conductivity change rate of graphene as a function of grain size change rate with a fit (blue curve), showing an exponential relationship. The thermal/electrical conductivity change rates of some typical metals (Au[47], Al[47], Ag[48] and Cu[49]) and semiconducting thermoelectric materials (BiSbTe[33], SrTe[34] and BiTeSe[35]) are also shown in different colours for comparison.

(BiTeSe, SrTe and BiSbTe)[33–35]. As shown in Fig. 5, the thermal conductivity change rate of graphene is much larger than those of all the compared materials, while the electrical conductivity change rate of graphene is larger than those of thermoelectric materials but smaller than those of metals. Moreover, the rates of change of electrical and thermal conductivity with grain size are almost the same for semiconducting thermoelectric materials. For instance, both the thermal and electrical conductivities of SrTe decrease by only 37% when its grain size is reduced by 99.8% of the pristine value ($\sim 500$ times difference)[34]. In contrast, when the grain size of graphene is decreased by 90% (10 times difference), its thermal and electrical conductivities are reduced by 89% (10 times difference) and 48% (two times difference), respectively. These results further confirm that nano-crystallization should be an efficient way to improve the thermoelectric properties of graphene.

The GBs in graphene can be approximated as linear periodic arrays of dislocations[12]. The crystal momentum conservation has a crucial role in the transmission of charge carriers across these topological defects[14]. As reported previously[14], these GBs can be classified into two classes according to the matching vectors $(n_L, m_L)$ and $(n_R, m_R)$ that belong to the left and right crystalline domains, respectively. If only one matching vector fulfills the criterion $(n-m) = 3q$ ($q$, integer), then the GB is of class-II type. Otherwise it belongs to class-I. For class-II GB, there is significant misalignment of the allowed momentum–energy manifolds corresponding to the two crystalline domains of graphene, which introduces a transport gap (usually 0.3–1.4 eV) that depends exclusively on the periodicity[14,45]. That is, class-II GB perfectly reflects low-energy carriers. In contrast, class-I GB is highly transparent with respect to charge carriers[14,45]. Different from the strong dependence of charge carrier transport on GB type, the phonon transmission shows a weak dependence on GB type[45]. More importantly, both types of GBs greatly suppress the phonon transmission[45]. Therefore, the thermal conductivity change rate of graphene as a function of grain size is dramatically larger than the electrical conductivity change rate. However, the deep mechanisms and physical pictures need to be further studied in the future.

In conclusion, we report a SACVD method to grow well-stitched high-quality monolayer graphene films with a tunable uniform grain size from $\sim 200$ nm to $\sim 1$ µm, by using a Pt substrate with medium carbon solubility. Using these

materials, we determined the scaling laws of the thermal and electrical conductivities of graphene films as a function of grain size. It was found that the thermal conductivity of polycrystalline graphene films dramatically decreases with decreasing grain size by a small thermal boundary conductance of $\sim 3.8 \times 10^9$ W m$^{-2}$ K$^{-1}$, while the electrical conductivity slowly decreases with an extraordinarily small GB transport gap of $\sim 0.01$ eV and GB resistivity of $\sim 0.3$ kΩ µm. Moreover, the changes in both the thermal and electrical conductivities with grain size change are greater than those of typical semiconducting thermoelectric materials. These findings provide valuable information for tuning the thermal and electrical properties of graphene for electronic, optoelectronic and thermoelectric applications through grain size engineering.

## Methods

**SACVD growth of polycrystalline graphene films.** A typical procedure for the SACVD growth of graphene films with grain sizes $<1$ µm includes four steps: surface growth, etching, segregation and surface growth. Before growth, a piece of Pt foil (180 µm thick, 99.9 wt% metal basis, 20 mm × 20 mm) was rinsed with acetone and ethanol in sequence for 1 h each, loaded into a fused-silica tube (inner diameter: 22 mm), heated to a certain temperature under the protection of hydrogen, and then annealed for 10 min to remove any residual carbon or organic substances. The first step: surface growth was started with the substrate being held for a certain time under a mixture of methane and hydrogen. In the second step the methane and hydrogen flows were turned off, and pure argon (700 sccm) was introduced to the system for 20 min to etch the graphene grown on the Pt in the first step into the bulk. In the third step, a small amount of hydrogen was introduced into the system to mix the argon flow, initiating segregation of the carbon to form small graphene domains on the Pt surface. In the fourth step a low flow rate of methane (0.1 sccm) was introduced into the system while maintaining the hydrogen and argon flows, to cause surface growth of the graphene domains produced in step 3 to form a continuous polycrystalline film. The detailed experimental conditions were given in the main text and Supplementary Information. Polycrystalline graphene films with grain sizes larger than 1 µm were grown by conventional CVD as previously reported[11].

**Structural characterization.** To investigate the structure of the graphene formed at different stages, the Pt foil was quickly pulled out of the high-temperature zone after SACVD growth. The furnace was then shut down and the methane flow was stopped after the furnace temperature had decreased to 600 °C. Finally, the Pt foil was taken out and characterized by SEM (Nova NanoSEM 430, acceleration voltage of 5 kV). The small graphene domains and polycrystalline films were transferred onto Si/SiO$_2$ (290 nm) substrates using a improved bubbling transfer method[11], in which the Poly(methyl methacrylate) (PMMA) used for transfer had a smaller molecular weight (600 kDa, 4 wt.% in ethyl lactate) and the acetone used for removing PMMA was heated at 50 °C to enhance the solubilities, for

morphological and quality analysis by optical microscopy (Nikon LV100D) and Raman spectroscopy (JY HR800, 532 nm laser wavelength, 1 μm spot size, 1 s integration time, laser power below 2 mW). The polycrystalline films were transferred to TEM grids by using a improved bubbling transfer method mentioned above for GB analysis by TEM (FEI Tecnai F20, 200 kV; FEI Tecnai T12, 120 kV; FEI Titan G2 equipped with an image-side spherical aberration corrector, 80–300 kV).

**Thermal and eletrical transport measurments.** We used a Renishaw inVia micro-Raman spectroscopy system with a 532 nm laser as excitation source to measure the thermal conductivity of the graphene films. A laser beam with a spot size of 1 μm was focused onto the samples through a × 50 objective (NA = 0.8), and the integration time at each position was 10 s. The temperature rise was determined from the shift of the Raman 2D peak. The sheet resistances of the graphene films were measured by a four-probe method (RTS-9) at room temperature. These two methods have been widely used in the literatures[12,13,38,44,50]. It is worth noting that the measured sheet resistance of the graphene on $SiO_2/Si$ substrate and thermal conductivity of the suspended graphene in our experiments show the similar values with those of graphene with similar grain size reported in the literatures[44,50]. Moreover, we also measured the thermal conductivity of the suspended mechanical exfoliated graphene films, which gives a value of up to $5.7 \times 10^3$ W m$^{-1}$ K$^{-1}$, close to the reported value ($5.3 \times 10^3$ W m$^{-1}$ K$^{-1}$) (ref. 38). These comparison results give concrete validations for our methodology.

**Data availability.** The data that support the findings of this study are available from the corresponding author upon request.

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

## Acknowledgements

This work was supported by the Ministry of Science and Technology of China (No. 2016YFA0200101), National Science Foundation of China (Nos. 51325205, 51290273, 51521091, 51172240 and 51222202), and Chinese Academy of Sciences (Nos KGZD-EW-303-1 and KGZD-EW-T06). This work partly used the resources of the Center of Electron Microscopy of Zhejiang University.

## Author contributions

W.R. proposed the project. W.R. and H.M.C. supervised the project. W.R. and T.M. designed the experiments. T.M. performed the experiments. Z.L. performed TEM measurements under the supervision of X.M. J.W. performed thermal measurements under the supervision of H.C. and N.X. X.R. performed aberration-corrected HRTEM measurements under the supervision of C.J. W.R. and T.M. analysed the experimental data. W.R., T.M. and H.M.C. wrote the manuscript. All the authors discussed the results and commented on the manuscript.

## Additional information

**Competing financial interests:** The authors declare no competing financial interests.

