## [Peer Review File · Nature Communications]

Reviewers' Comments:

Reviewer #1 (Remarks to the Author)

The authors synthesized polycrystalline graphene with the ASCVD method. They claimed that grain size can be controlled, and thus tailored, for controlling the thermal and electrical conductivities of graphene, which vary sharply with varying grain sizes. Given this control and possible thermoelectric applications, polycrystalline graphene is superior to some typical metals (Au, Ag, Cu, and Al) and semiconducting thermoelectric materials (BiTeSe, SrTe, and BiSbTe), and thus can be exploited for thermoelectric applications.

The major novelty claimed by the author is the controlled synthesis of polycrystalline graphene with 200 nm to 1 micron grain sizes. The synthesis is indeed interesting, but does not represent a major breakthrough as there exist many publications on synthesis. A major drawback of the "expansion and stitching" method is that GBs formed this way are not natural, so more defects and microstructures exist at such GBs. The dependences of thermal and electrical conductivities on grain sizes are well expected, although the authors did provide some new measurements. Such quantitative dependences are good to know about graphene as an interesting material, but its extended, thermoelectric applications, are still speculation compared to conventional semiconducting thermoelectric materials (BiTeSe, SrTe, and BiSbTe), since grain size dependence is just one of many parameters which determine the performances.

Some specific questions:

1. Fig. 1e: could the authors verify that the graphene samples are indeed monolayered.
2. Are there any surface roughness, and buckling or overlapping at the grain boundaries, since they are "sintered" rather than natural?
3. While the grain size may be "well controlled," the grain rotations are not so as seen from Fig. 2. Thermal /electrical conductivities may well depend on orientations. It is indeed difficult characterization, but current study lacks sufficient statistics to make a claim about the relation between grain size and misorientation.
4. The authors mentioned that "The relative grain rotation angle distribution becomes broad with increasing grain size." In other words, the GB orientation becomes more random with increasing grain size. Meanwhile, GB orientation is in a narrow range of 0-10 degrees in small grain size graphene. A possible reason is that grain orientation and grain size are coupled during synthesis. More evidence should be supplied to support the authors' claim that "... thermal conductivity is independent of GB orientation" (in the abstract).
5. How does high temperature affect GB structure and introduce more defects?
6. Thermal and electrical conductivity measurements on graphene are nontrivial, no matter what methods are used. Could the authors supply some concrete validations for their methodology?

Reviewer #2 (Remarks to the Author)

The present manuscript focuses on experiments that demonstrate dependency of thermal and electrical conductivity of graphene films on the domain size. Control of the domain size is achieved by appropriate change of CVD conditions. Graphene is grown on Pt which facilitates growth of graphene with domain sizes in the submicron range. The choice of techniques to evaluate thermal and electrical conductivity of graphene films are quite standard.

While the scope of the presented work is of interest, the experiments are well thought out and the data are well presented, the authors apparently overlooked very similar previous studies on this subject. There is a particularly significant overlap between the presented study and the recently published paper "Electrical and thermal conductivity of low temperature CVD graphene: the effect of disorder" by Vlasiouk et al. 2011 Nanotechnology 22 275716 <http://dx.doi.org/10.1088/0957-4484/22/27/275716>

Taking into account the low level novelty, and the fact the manuscript was written without analyzing highly relevant previously published results, I do not recommend its publication in the present form. A major revision might make this manuscript suitable for a publication in a more specialized journal.

Reviewer #3 (Remarks to the Author)

The work by Ma et al. describes the growth and characterization of CVD graphene films with tunable grain size. By using a novel adsorption technique on Pt, the authors can grow polycrystalline samples with good control over average grain size and size distribution. This allows for a correlation of grain size with thermal and electrical measurements to determine the effects of grain boundaries. While the results are not surprising, I find the work to be carefully done overall.

There are two conclusions that I find unjustified. It is claimed from Fig. S7 that the graphene grains are "perfectly stitched." It is not just the physical connectivity of grains that are important, but rather the size of the disorder region that impacts grain boundary performance. This type of imaging tells us nothing in that regard.

The variation in the electrical conductivity of large grain samples is concluded to mean that grain boundary resistance "strongly depends" on relative orientation. There are various factors that contribute to enhanced boundary resistivity (doping, size of disordering region, etc.). Since the authors do not control for these properties, it is hard to pinpoint grain orientation as the cause.

Responses to the Reviewers' Comments

Reviewer #1 (Remarks to the Author):

The authors synthesized polycrystalline graphene with the ASCVD method. They claimed that grain size can be controlled, and thus tailored, for controlling the thermal and electrical conductivities of graphene, which vary sharply with varying grain sizes. Given this control and possible thermoelectric applications, polycrystalline graphene is superior to some typical metals (Au, Ag, Cu, and Al) and semiconducting thermoelectric materials (BiTeSe, SrTe, and BiSbTe), and thus can be exploited for thermoelectric applications.

The major novelty claimed by the author is the controlled synthesis of polycrystalline graphene with 200 nm to 1 micron grain sizes. The synthesis is indeed interesting, but does not represent a major breakthrough as there exist many publications on synthesis. A major drawback of the "expansion and stitching" method is that GBs formed this way are not natural, so more defects and microstructures exist at such GBs. The dependences of thermal and electrical conductivities on grain sizes are well expected, although the authors did provide some new measurements. Such quantitative dependences are good to know about graphene as an interesting material, but its extended, thermoelectric applications, are still speculation compared to conventional semiconducting thermoelectric materials (BiTeSe, SrTe, and BiSbTe), since grain size dependence is just one of many parameters which determine the performances.

Response: We thank the reviewer very much for the kind comments.

We agree with the reviewer that many publications on the CVD growth of graphene have been reported. However, to the best of our knowledge, nearly all the publications on the grain control focus on the growth of large-scale single-crystal graphene, which is expected to have promising applications in electronics and optoelectronics. Actually, polycrystalline graphene has also been predicted and/or demonstrated to have many fascinating properties and applications that are different from single-crystal graphene because of the presence of grain boundaries, such as chemical sensing, valley polarization, and plasmonics¹⁻⁴. Therefore, controlled growth of graphene with very small grain size is not only fundamental but also technologically important in order to tune the properties of graphene for various applications.

However, from the point of view of crystal growth, it is equally difficult to reduce and increase nucleation density of graphene on metals by CVD to fabricate large-size single-crystal graphene and polycrystalline graphene with nano-sized grains, respectively, while keeping monolayer growth. For surface adsorption growth on the commonly used Cu with a low carbon solubility, a high-concentration carbon source and/or defective substrates have usually been used to obtain a high domain density, however, these conditions led to the formation of multi-layer graphene domains⁵⁻⁷. It is well known that the graphene films segregated from Ni with a high carbon solubility are usually nonuniform multi-layers⁸⁻¹⁰. As a result, the polycrystalline graphene films prepared so far based on either surface absorption or segregation mechanism usually have a grain size ranging from $\sim 1 \mu\text{m}$ to $\sim 1 \text{mm}$ ^{4, 5, 11-19}, which is larger than the electron and phonon mean free paths (a few hundreds of nanometers)²⁰,

and/or have very broad grain size distributions^{12, 13} For example, although Huang et al¹² demonstrated the growth of graphene films with an average grain size of ~250 nm by using a low-purity Cu substrate and high CH₄ flow rate, the grain size distribution is very broad, ranging from a few nanometers to over 1 micrometer.

In all, the fabrication of monolayer polycrystalline graphene films with grain sizes that are smaller than the electron and phonon mean free paths and have a narrow distribution still remains a great challenge. This strongly hinders the experimental studies on the real influence of grain size on the electrical and thermal transport properties of graphene films.

Table 1 Synthesis method and the grain size of the polycrystalline graphene films obtained.

Substrate	Synthesis Method	Grain Size	Reference
Cu	High CH ₄ flow rate and high pressure	6 μm	Nano Lett., 2010 ¹⁶
Cu	Increase the flow rate ratio of CH ₄ to H ₂ .	3 μm	Adv. Mater., 2011 ⁶
Cu	High CH ₄ flow rate	4 μm	Nano Lett., 2011 ⁷
Cu	Increase the flow rate ratio of CH ₄ to H ₂ .	20 μm	Nat. Mater., 2011 ¹¹
Cu	Use low-purity copper (99.8%) and high CH ₄ flow rate	250 nm	Nature, 2011 ¹²
Cu	Use low-purity copper (99.8%)	5 μm	ACS Nano, 2011 ¹³
Cu	Growth inside a copper-foil enclosure	100 μm	Nano Lett., 2012 ¹⁴
Cu	High CH ₄ flow rates	1 μm	Science, 2012 ¹⁵
Pt	Low temperature, high CH ₄ flow rate	30 μm	Nat. Comm., 2012 ¹⁶
Cu	Tuning the reactor pressure	50 μm	Nat. Comm., 2013 ⁵

Cu	Low temperature, plasma-enhanced CVD	5 μm	Nat. Comm., 2015 ¹⁹
----	--------------------------------------	-----------------	-----------------------------------

In the present work, we develop a segregation-adsorption CVD (SACVD) method to greatly increase the nucleation density of graphene (by segregation) and keep monolayer growth (by surface adsorption) simultaneously, by using a Pt substrate with medium carbon solubility. As a result, we are able to grow monolayer graphene films with a tunable grain size from ~ 200 nm to ~ 1 μm . In particular, the grain size distributions of all samples are very narrow, with a variation within 400 nm. According to the reviewer's suggestion, we have also performed a great number of atomic-resolution TEM measurements to identify the detailed microstructure of the grain boundaries. It was found that the neighboring grains are perfectly stitched together without defects, buckling, overlapping and microstructures existing at the grain boundaries (Fig. R1 in this response). The synthesis of well-stitched high-quality monolayer graphene films with a tunable uniform grain size that are smaller than the electron and phonon mean free paths is one novelty of this work.

Figure R1. HRTEM images of the graphene films with grain size of ~ 200 nm (**a,c**) and ~ 700 nm (**b,d**). The pentagons (blue), heptagons (red) and hexagons (yellow) in the grain boundaries are outlined. All images were processed with an improved Wiener-Filtering to remove the noises. The scale bars are 1 nm.

Indeed, the influences of grain boundaries on the electrical and thermal conductivity have been widely studied theoretically and experimentally. However, the theoretical results are usually in conflict with each other mainly because of the use of different models. Similarly, the experimental results are also inconsistent with each other because of the use of graphene films that are lack of good control on the grain size. Moreover, it is worth noting that the graphene films used in the previous studies usually have a grain size larger than $1 \mu\text{m}$, however, the contributions to electrical and thermal transports due to scattering from grain boundaries are more significant when the grain size is smaller than the phonon and electron mean free paths (\sim a few hundreds of nanometers). Therefore, it is actually still ambiguous about the real

influence of grain size on the electrical and thermal conductivities of graphene.

In the present study, using graphene with tunable uniform small grain size prepared by our SACVD method, we give clear experimental evidence and scaling laws of the influence of grain size on the electrical and thermal conductivities of graphene films on a large scale. Moreover, it is worth noting that we have compared the influence of grain size on the electrical and thermal transport properties by measuring the samples with the same grain size. We found that the thermal conductivity of graphene films dramatically decreases with decreasing grain size by a small thermal boundary conductance of $\sim 3.8 \times 10^9 \text{ Wm}^{-2}\text{K}^{-1}$, while the electrical conductivity slowly decreases with an extraordinarily small grain boundary transport gap of $\sim 0.01 \text{ eV}$. Furthermore, the changes in both the thermal and electrical conductivities with grain size change are greater than those of typical semiconducting thermoelectric materials. These findings are another novelty of our work, which provide valuable information for tuning the thermal and electrical properties of graphene for electronic, optoelectronic and thermoelectric applications through grain size engineering.

Based on the above, we believe that the present work represents a significant advance not only in the controlled growth of graphene by CVD but also in the fundamental understanding on the intrinsic influence of grain size on the electrical and thermal conductivity of graphene.

Some specific questions:

1. Fig. 1e: could the authors verify that the graphene samples are indeed monolayered.

Response: We thank the reviewer very much for kind suggestion.

In order to confirm that the graphene obtained by our SACVD method is indeed monolayer, we have performed optical, Raman and HRTEM measurements on two typical graphene films with grain size of 200 nm and 1 μm (Figs. R2 and R3). It can be seen from the optical image that the graphene films have a uniform contrast, indicating the uniform number of layers. Both the Raman spectra taken from more than 50 random positions and the Raman mapping of $\sim 20 \times 30 \mu\text{m}^2$ show an intensity ratio of 2D peak to G peak of ~ 2 , a fingerprint of monolayer graphene. HRTEM investigations on the edges of the films further confirm that they are indeed monolayers.

Figures R2 and R3 and the related discussions have been given in the revised manuscript.

Figure R2. Structure characterization on the graphene films with ~200 nm-sized grains. (a) Raman spectra randomly acquired from more than 50 positions. (b) Optical image of the graphene film transferred onto a SiO₂/Si substrate. (c) HRTEM image of the edge of the graphene film. (d,e) Raman mappings of the intensity ratios of 2D peak to G peak (d) and D peak to G peak (e).

Figure R3. Structure characterization on the graphene films with $\sim 1 \mu\text{m}$ -sized grains. (a) Raman spectra randomly acquired from more than 50 positions. (b) Optical image of the graphene film transferred onto a SiO_2/Si substrate. (c) HRTEM image of the edge of the graphene film. (d,e) Raman mappings of the intensity ratios of 2D peak to G peak (d) and D peak to G peak (e).

2. Are there any surface roughness, and buckling or overlapping at the grain boundaries, since they are "sintered" rather than natural?

Response: We thank the reviewer very much for kind comments.

In our SACVD method, we used a very low flow rate of methane (0.1 sccm) to induce surface growth of the graphene domains that are pre-formed by segregation. The resulting slow growth rate facilitates the relaxation of metal-carbon system

towards thermal equilibrium during growth, and consequently enables the merging of graphene domains naturally. In order to obtain atomic-resolution information of the grain boundaries, we have performed aberration-corrected HRTEM measurements, where 80 kV electron beam was used to minimize the irradiation damage to the graphene membranes. Figure R1a and c show typical HRTEM images of the graphene film with a grain size of ~200 nm. It can be found that two grains are stitched together by pentagons and heptagons without any other defects or microstructures. Similarly, the neighboring grains are also perfectly stitched together only by pentagons and heptagons in the graphene films with grain size of ~700 nm (Fig. R1b and d). Extensive measurements on the graphene films with different grain size gave similar results. In all the investigate samples, the grain boundaries exhibit atomically sharp interface regions by chains of pentagons and heptagons embedded in the hexagonal lattice of graphene without overlapping, buckling and other defects.

Figure R1 and the related discussions have been added in the revised manuscript.

3. While the grain size may be "well controlled," the grain rotations are not so as seen from Fig. 2. Thermal/electrical conductivities may well depend on orientations. It is indeed difficult characterization, but current study lacks sufficient statistics to make a claim about the relation between grain size and misorientation.

Response: We thank the reviewer for the comment. In order to identify the grain orientation, we have performed electron diffraction measurements on the graphene films with different grain size, in which each diffraction pattern was captured using a

~1 μm diameter aperture to exclude the Cu grid support. As shown in Supplementary Fig. S9, the graphene films with a grain size of ~200 nm exhibit a narrow grain orientation distribution, while the graphene films with a grain size of ~500 and 700 nm have much broader grain orientation distributions. We have measured more than 20 areas for each sample, which give similar results. Nevertheless, as the reviewer mentioned, it still lacks sufficient statistics to make a claim about the relation between grain size and misorientation. Therefore, we have deleted all the discussions about the grain orientations in the revised manuscript.

4. The authors mentioned that "The relative grain rotation angle distribution becomes broad with increasing grain size." In other words, the GB orientation becomes more random with increasing grain size. Meanwhile, GB orientation is in a narrow range of 0-10 degrees in small grain size graphene. A possible reason is that grain orientation and grain size are coupled during synthesis. More evidence should be supplied to support the authors' claim that "... thermal conductivity is independent of GB orientation" (in the abstract).

Response: We thank the reviewer for the kind suggestion. We agree with the reviewer that it is possible that grain orientation and size are coupled during the synthesis. According to the reviewer's suggestion, we have tried to synthesize graphene films with the same grain size but different grain orientations to provide more evidence on the influences of grain orientation on the thermal and electrical conductivities, but failed. Based on the present data, however, it is indeed difficult to identify whether the

thermal/electrical conductivities depend on orientations. Therefore, we have deleted all the grain orientation related statements.

Instead, we extracted the scaling law of the thermal conductivity of graphene as a function of grain size based on kinetic theory of phonon transport. This gives clear evidence that the thermal conductivity of graphene films dramatically decreases with decreasing grain size by a small thermal boundary conductance of $\sim 3.8 \times 10^9 \text{ Wm}^{-2}\text{K}^{-1}$, which are perfectly consistent with the theoretical predictions. Using this scaling law, we also estimated that the critical size of grains below which the contribution from the grain boundaries becomes comparable to the scattering from the grain is $\sim 1.4 \text{ }\mu\text{m}$.

5. How does high temperature affect GB structure and introduce more defects?

Response: The grain boundary in graphene is a topological defect, which is formed by chains of alternating pentagons and heptagons embedded in the hexagonal lattice of graphene⁴. Compared to the carbon atoms within grain, the carbon atoms in the grain boundaries have much higher activity. At high temperatures, the enhanced thermal vibration of carbon atoms makes the chemical bonds more easily reach the critical bond length and break²¹. In situ aberration corrected TEM studies also show that high temperature can accelerate dynamic evolution of the graphene boundaries, increasing bond rotation and atomic addition/loss²².

In the Raman spectra of graphene, the G peak is related to the in-plane bond-stretching motion of pairs of sp^2 carbon atoms²³. This mode does not require the presence of sixfold rings, and so it occurs at all sp^2 sites, not only those in rings²³. The

D mode is forbidden in perfect graphene only becomes active in the presence of disorders or defects²³. Its intensity is strictly connected to the presence of sixfold aromatic rings²³. The intensity ratio of D peak to G peak (I_D/I_G) represents the size of ordered sp^2 aromatic ring clusters/domains in a carbon atom network consisting of ordered and disordered clusters/domains. To understand the influence of temperature on the grain boundary structure, we plotted the G peak intensity and I_D/I_G as a function of laser power (Fig. R4 in this response). If no defects are generated, the intensities of both G peak and D peak should increase with increasing laser power but keeping a constant I_D/I_G . However, it is worth noting that when the laser power is larger than 1.2 eV, the G peak intensity decreases along with an increase of D peak intensity and I_D/I_G (blue curve in Fig. R4). Moreover, both the G peak intensity and I_D/I_G cannot recover their original values at the same laser power when the laser power is decreased (brown curve in Fig. R4). Combined with the high activity of the carbon atoms at grain boundaries, we suggest that the high temperature (~650 K) leads to the breaking of graphene at the grain boundaries due to strong thermal vibration.

Figure R4 and the above related discussions have been given in the revised manuscript.

Figure R4. Intensity of the G peak (a) and I_D/I_G (b) as a function of laser power.

6. Thermal and electrical conductivity measurements on graphene are nontrivial, no matter what methods are used. Could the authors supply some concrete validations for their methodology?

Response: We thank the reviewer for the advices. In our experiments, we used four-probe method and confocal micro-Raman spectroscopy to measure the sheet resistance and thermal conductivity of the graphene, respectively. These two methods have been widely used in the literature^{4,18,24-26}. It is worth noting that the measured sheet resistance of the graphene on SiO₂/Si substrate and thermal conductivity of the suspended graphene show the similar values with those of graphene with similar grain size reported in the literature^{24,25}. Moreover, we also measured the thermal conductivity of the suspended mechanical exfoliated graphene films, which gives a value of up to $5.7 \times 10^3 \text{ Wm}^{-1}\text{K}^{-1}$, close to the reported value ($5.3 \times 10^3 \text{ Wm}^{-1}\text{K}^{-1}$)²⁶. These comparison results give concrete validations for our methodology.

The above discussions have been added in the revised manuscript.

Reviewer #2 (Remarks to the Author):

The present manuscript focuses on experiments that demonstrate dependency of thermal and electrical conductivity of graphene films on the domain size. Control of the domain size is achieved by appropriate change of CVD conditions. Graphene is grown on Pt which facilitates growth of graphene with domain sizes in the submicron range. The choice of techniques to evaluate thermal and electrical conductivity of graphene films are quite standard.

While the scope of the presented work is of interest, the experiments are well thought out and the data are well presented, the authors apparently overlooked very similar previous studies on this subject. There is a particularly significant overlap between the presented study and the recently published paper "Electrical and thermal conductivity of low temperature CVD graphene: the effect of disorder" by Vlassiouk et al. 2011 Nanotechnology 22 275716 <http://dx.doi.org/10.1088/0957-4484/22/27/275716>

Taking into account the low level novelty, and the fact the manuscript was written without analyzing highly relevant previously published results, I do not recommend its publication in the present form. A major revision might make this manuscript suitable for a publication in a more specialized journal.

Response: We thank the reviewer very much for the comments and suggestions.

In the Nanotechnology paper, the authors fabricated graphene materials with different extent of disorder by low-temperature CVD on Cu and Ni substrates, and investigated the thermal/electrical properties as a function of domain size which was

extracted based on the intensity ratio of Raman D peak to G peak (I_D/I_G). However, this is totally different from what we presented in our manuscript. The D mode is forbidden in perfect graphene only becomes active in the presence of disorders or defects, and its intensity is strictly connected to the presence of sixfold aromatic rings²³. It is well known that many kinds of disorders and defects can activate D peak, such as Stone-Wales defects, vacancies, holes, grain boundaries, edges, distortions, and contaminations. All these disorders are very common in the graphene prepared by low-temperature CVD. Moreover, as shown in our manuscript, the well-stitched grain boundaries do not give rise to visible D peak. Actually, the I_D/I_G represents the size of ordered sp^2 aromatic ring clusters/domains in a carbon network consisting of ordered and disordered clusters/domains rather than the grain size^{23,27}. Therefore, it is impossible to give a clear relationship between thermal/electrical properties and grain size except for the influence of the extent of disorder in the Nanotechnology paper.

In contrast, we developed a SACVD method to greatly increase the nucleation density of graphene (by segregation) and keep monolayer growth (by surface adsorption) simultaneously, by using a Pt substrate with medium carbon solubility. As a result, we are able to grow high-quality monolayer graphene films with a tunable uniform grain size from ~200 nm to ~1 μm , which have never been achieved before by CVD based on either surface adsorption or segregation mechanism. Moreover, as confirmed by extensive HRTEM measurements, the grain boundaries are well-stitched and exhibit atomically sharp interface regions by chains of pentagons and heptagons embedded in the hexagonal lattice of graphene without overlapping, buckling and

other defects (Fig. R1 in this response). The successful synthesis of these graphene films allows us to investigate the real influence of grain size on the electrical and thermal transport properties of graphene films on a large scale.

As a result, different scaling laws have been extracted in these two studies. Our results show that the thermal conductivity of graphene films decreases more significantly than the electrical conductivity with decreasing grain size, while the Nanotechnology paper shows that the thermal conductivity decreased much slower than the electrical conductivity with decreasing domain size (i.e., increasing degree of disorder). Moreover, we have extracted the thermal boundary conductance and grain boundary electrical transport gap, and found that the changes in both the thermal and electrical conductivities with grain size change are greater than those of typical semiconducting thermoelectric materials. Our findings provide valuable information for tuning the thermal and electrical properties of graphene for electronic, optoelectronic and thermoelectric applications through grain size engineering.

Based on the above, we believe that the present work has a high novelty and represents a significant advance not only in the controlled growth of graphene by CVD but also in the fundamental understanding on the intrinsic influence of grain size on the electrical and thermal conductivity of graphene.

In addition, we have cited the Nanotechnology paper in our revised manuscript.

Reviewer #3 (Remarks to the Author):

The work by Ma et al. describes the growth and characterization of CVD graphene

films with tunable grain size. By using a novel adsorption technique on Pt, the authors can grow polycrystalline samples with good control over average grain size and size distribution. This allows for a correlation of grain size with thermal and electrical measurements to determine the effects of grain boundaries. While the results are not surprising, I find the work to be carefully done overall.

There are two conclusions that I find unjustified. It is claimed from Fig. S7 that the graphene grains are "perfectly stitched." It is not just the physical connectivity of grains that are important, but rather the size of the disorder region that impacts grain boundary performance. This type of imaging tells us nothing in that regard.

Response: We thank the reviewer very much for the kind comments.

According to the reviewer's suggestion, we have performed aberration-corrected HRTEM measurements to obtain atomic-resolution information of the grain boundaries, where 80 kV electron beam was used to minimize the irradiation damage to the graphene membranes. Figure R1a and c show typical HRTEM images of graphene film with a grain size of ~200 nm. It can be found that two grains are stitched together by pentagons and heptagons without any other defects or microstructures. Similarly, the neighboring grains are also perfectly stitched together only by pentagons and heptagons in the graphene films with grain size of ~700 nm (Fig. R1b and d). Extensive measurements on the graphene films with different grain size gave similar results. In all the investigate samples, the grain boundaries exhibit atomically sharp interface regions by chains of pentagons and heptagons embedded in the hexagonal lattice of graphene without overlapping, buckling and other defects.

Figure R1 and the related discussions have been added in the revised manuscript

Figure R1. HRTEM images of the graphene films with grain size of ~ 200 nm (a,c) and ~ 700 nm (b,d). The pentagons (blue), heptagons (red) and hexagons (yellow) in the grain boundaries are outlined. All images were processed with an improved Wiener-Filtering to remove the noises. The scale bars are 1 nm.

The variation in the electrical conductivity of large grain samples is concluded to mean that grain boundary resistance "strongly depends" on relative orientation. There are various factors that contribute to enhanced boundary resistivity (doping, size of disordering region, etc.). Since the authors do not control for these properties, it is hard to pinpoint grain orientation as the cause.

Response: We agree with the reviewer that there are various factors contributing to enhanced boundary resistivity. Therefore, it is hard to identify whether the variation in the electrical conductivity is attributed to the influence of grain orientation, and we

have deleted the orientation related statements in our revised manuscript. Instead, we further analyzed the electrical transport data. We fit the electrical conductivity data using modified Arrhenius equation²⁸ $\sigma = \sigma_0 \exp\{-Ea/[RT(l_g+c)]\}$ (Fig. 4d), where σ is the electrical conductivity of the polycrystalline graphene films, σ_0 is the electrical conductivity within the grain, Ea is the GB transport gap, R is the universal gas constant, T is the absolute temperature, l_g is the grain size, and c is the correction value. The fitting gives $\sigma_0 \approx 2.85 \times 10^6 \text{ S m}^{-1}$ and $Ea \approx 0.01 \text{ eV}$. The GB transport gap extracted here is smaller than the theoretically predicted value²⁹, further confirming the perfect stitching of neighboring grains in our graphene films. Using this scaling law, we found that the GBs begin to dominant the electrical conductivity of the polycrystalline graphene films only when the grain size is smaller than $l_g \approx 0.8 \mu\text{m}$.

Moreover, in sharp contrast to thermal conductivity, the electrical conductivity of the graphene films shows a much weaker dependence on grain size. As shown in Supplementary Fig. S15, when the mean grain size is increased from $\sim 200 \text{ nm}$ to $\sim 1 \text{ mm}$ (five orders of magnitude increase), there is only a 4-fold increase in electrical conductivity. The above results suggest that increasing grain size is not an efficient way to improve the electrical conductivity of graphene for transparent conductive electrode applications when the grain size is larger than $1 \mu\text{m}$. However, together with the strong dependence of thermal conductivity on grain size, it is reasonable to expect that graphene has a potential for thermoelectric applications by grain nano-crystallization. For instance, the thermal conductivity of graphene films with a grain size of 1 nm is extrapolated to be $\sim 3.8 \text{ Wm}^{-1}\text{K}^{-1}$, a $\sim 1,400$ times decrease

compared to single-crystal graphene. However, the electrical conductivity is extrapolated to be $\sim 5.8 \times 10^5 \text{ S m}^{-1}$, only a ~ 10 times decrease compared to graphene with a millimeter grain size.

Reference

1. Yasaei, P. *et al.* Chemical sensing with switchable transport channels in graphene grain boundaries. *Nat. Commun.* **5**, 4911 (2014).
2. Lahiri, J., Lin, Y., Bozkurt, P., Oleynik, I. I. & Batzill, M. An extended defect in graphene as a metallic wire. *Nat. Nanotechnol.* **5**, 326-329 (2010).
3. Fei, Z. *et al.* Electronic and plasmonic phenomena at graphene grain boundaries. *Nat. Nanotechnol.* **8**, 821-825 (2013).
4. Yazyev, O. V. & Chen, Y. P. Polycrystalline graphene and other two-dimensional materials. *Nat. Nanotechnol.* **9**, 755-767 (2014).
5. Zhou, H. *et al.* Chemical vapour deposition growth of large single crystals of monolayer and bilayer graphene. *Nat. Commun.* **4**, 2096 (2013).
6. Wu, B. *et al.* Equiangular hexagon-shape-controlled synthesis of graphene on copper surface. *Adv. Mater.* **23**, 3522-3525 (2011).
7. Robertson, A. W. & Warner, J. H. Hexagonal single crystal domains of few-layer graphene on copper foils. *Nano Lett.* **11**, 1182-1189 (2011).
8. Li, X., Cai, W., Colombo, L. & Ruoff, R. S. Evolution of graphene growth on Ni and Cu by carbon isotope labeling. *Nano Lett.* **9**, 4268-4272 (2009).
9. Kim, K. S. *et al.* Large-scale pattern growth of graphene films for stretchable

- transparent electrodes. *Nature* **457**, 706-710 (2009).
10. Reina, A. *et al.* Large area, few-layer graphene films on arbitrary substrates by chemical vapor deposition. *Nano Lett.* **9**, 30-35 (2009).
 11. Yu, Q. *et al.* Control and characterization of individual grains and grain boundaries in graphene grown by chemical vapour deposition. *Nat. Mater.* **10**, 443-449 (2011).
 12. Huang, P. Y. *et al.* Grains and grain boundaries in single-layer graphene atomic patchwork quilts. *Nature* **469**, 389-392 (2011).
 13. Kim, K. *et al.* Grain boundary mapping in polycrystalline graphene. *ACS Nano* **5**, 2142-2146 (2011).
 14. Petrone, N. *et al.* Chemical vapor deposition-derived graphene with electrical performance of exfoliated graphene. *Nano Lett.* **12**, 2751-2756 (2012).
 15. Tsen, A. W. *et al.* Tailoring electrical transport across grain boundaries in polycrystalline graphene. *Science* **336**, 1143-1146 (2012).
 16. Li, X. *et al.* Graphene films with large domain size by a two-step chemical vapor deposition process. *Nano Lett.* **10**, 4328-4334 (2010).
 17. Gao, L. *et al.* Repeated growth and bubbling transfer of graphene with millimetre-size single-crystal grains using platinum. *Nat. Commun.* **3**, 699 (2012).
 18. Cummings, A. W. *et al.* Charge transport in polycrystalline graphene: challenges and opportunities. *Adv. Mater.* **26**, 5079-5094 (2014).
 19. Boyd, D. A. *et al.* Single-step deposition of high-mobility graphene at reduced

- temperatures. *Nat. Commun.* **6**, 6620 (2015).
20. Fugallo, G. *et al.* Thermal conductivity of graphene and graphite: collective excitations and mean free paths. *Nano Lett.* **14**, 6109-6114 (2014).
 21. Yang, Z. *et al.* Temperature and strain-rate effects on the deformation behaviors of nano-crystalline graphene sheets. *Eur. Phys. J. B* **88**, 1-8 (2015).
 22. Gong, C., He, K., Chen, Q., Robertson, A. W. & Warner, J. H. In situ high temperature atomic level studies of large closed grain boundary loops in graphene. *ACS Nano* **10**, 9165-9173 (2016).
 23. Ferrari, A. C. & Robertson, J. Interpretation of Raman spectra of disordered and amorphous carbon. *Phys. Rev. B* **61**, 14095-14107 (2000).
 24. Cai, W. *et al.* Thermal transport in suspended and supported monolayer graphene grown by chemical vapor deposition. *Nano Lett.* **10**, 1645-1651 (2010).
 25. Duong, D. L. *et al.* Probing graphene grain boundaries with optical microscopy. *Nature* **490**, 235-239 (2012).
 26. Balandin, A. A. *et al.* Superior thermal conductivity of single-layer graphene. *Nano Lett.* **8**, 902-907 (2008).
 27. Ferrari, A. C. & Basko, D. M. Raman spectroscopy as a versatile tool for studying the properties of graphene. *Nat. Nanotechnol.* **8**, 235-246 (2013).
 28. Syed, R., Gavin, D. L. & Moynihan, C. T. Functional form of the Arrhenius equation for electrical conductivity of glass. *J. Am. Ceram. Soc.* **65**, c129-c130 (1982).

29. Yazyev, O. V. & Louie, S. G. Electronic transport in polycrystalline graphene.
Nat. Mater. **9**, 806-809 (2010).

Reviewers' comments:

Reviewer #1 (Remarks to the Author):

Two key clarifications are still needed before I am convinced of its eligibility for publication.

1. The major novelty claimed is the controlled synthesis (via expansion and stitching) of polycrystalline graphene. The authors did show an HRTEM image (Figure R1 or S11) trying to prove that these stitching grain boundaries are "perfectly stitched together without defects, buckling, overlapping and microstructures existing at the grain boundaries." But, we still need a larger TEM image (at least including about 10 grains) to demonstrate this claim is valid in general. As we know, stitching is always inferior to natural grain boundaries.

2. The authors mentioned "the thermal conductivity change rate of graphene is dramatically larger than the electrical conductivity change rate." It is highly desirable to further clarify and unify the mechanisms and physical pictures on how the grain size tunes thermal and electrical conductivities. By comparing thermal/electrical conductivity change rates of graphene with those of some typical metals and semiconducting thermoelectric materials, the authors tried to make a point that "graphene has a potential for thermoelectric applications by grain nano-crystallization." However, solid evidences are still needed to make the point convincing.

Reviewer #2 (Remarks to the Author):

After carefully evaluating the responses provided by the authors, I believe the changes in the revised manuscript have addressed all the major issues raised by the reviewers. In my opinion the revised manuscript can be published in its present form.

Reviewer #3 (Remarks to the Author):

The authors have satisfactorily addressed my concerns.

The only additional suggestion I have is the following. I'm not sure how to interpret the transport gap that is extracted. I assume it sets some energy scale for GB scattering. A more useful quantity would be GB resistivity, which the authors can probably determine through some simple modeling. I would like to see a comparison of this number against previous works (Huang, Yu, Tsen, etc.). I recommend publication after this addition.

Responses to the Reviewers' Comments

Reviewer #1 (Remarks to the Author):

Two key clarifications are still needed before I am convinced of its eligibility for publication.

1. The major novelty claimed is the controlled synthesis (via expansion and stitching) of polycrystalline graphene. The authors did show an HRTEM image (Figure R1 or S11) trying to prove that these stitching grain boundaries are “perfectly stitched together without defects, buckling, overlapping and microstructures existing at the grain boundaries.” But, we still need a larger TEM image (at least including about 10 grains) to demonstrate this claim is valid in general. As we know, stitching is always inferior to natural grain boundaries.

Response: We thank the reviewer very much for kind suggestion.

According to the reviewer's suggestion, we have tried to take large-area HRTEM images including about 10 grains. However, if the grain boundaries (GBs) can be clearly seen in HRTEM, the scope of the image should be no larger than $\sim 50 \times 50 \text{ nm}^2$. Since the smallest grain size of our polycrystalline graphene films is about 200 nm, it is impossible to have about 10 grains in a single image with atomic resolution. Figure R1 shows 4 large-area ($\sim 40 \times 40 \text{ nm}^2$) atomic resolution HRTEM images of graphene films with 200 nm-sized and 700 nm-sized grains, which contain one or two GBs in each image. These images further prove that the neighboring grains are perfectly stitched together without buckling, overlapping and microstructures. The contaminations on the surface of graphene films are PMMA residues, which are very

common for the graphene transferred by PMMA. The holes were generated by long-time electron beam irradiation during HRTEM observations.

It is worth noting that, in our SACVD method, we used a very low flow rate of methane to induce surface catalytic growth of the graphene domains. The resulting slow growth rate facilitates the relaxation of metal-carbon system towards thermal equilibrium during growth, which we believe is an important reason for the formation of well-stitched GBs.

We have added the large-area atomic resolution HRTEM images in the revised manuscript.

Figure R1. Large-area atomic resolution HRTEM images of the graphene films with grain size of ~200 nm (a,b) and ~700 nm (c,d). The insets are the corresponding fast Fourier transformation patterns, showing polycrystalline structure.

2. The authors mentioned “the thermal conductivity change rate of graphene is dramatically larger than the electrical conductivity change rate.” It is highly desirable to further clarify and unify the mechanisms and physical pictures on how the grain size tunes thermal and electrical conductivities. By comparing thermal/electrical conductivity change rates of graphene with those of some typical metals and semiconducting thermoelectric materials, the authors tried to make a point that “graphene has a potential for thermoelectric applications by grain nano-crystallization.” However, solid evidences are still needed to make the point convincing.

Response: We thank the reviewer very much for kind suggestions.

The GBs in graphene can be approximated as linear periodic arrays of dislocations¹. The crystal momentum conservation plays a crucial role in the transmission of charge carriers across these topological defects². As reported previously², these GBs can be classified into two classes according to the matching vectors (n_L, m_L) and (n_R, m_R) that belong to the left and right crystalline domains, respectively. If only one matching vector fulfills the criterion $(n - m) = 3q$ (q , integer), then the GB is of class-II type. Otherwise it belongs to class-I. For class-II GB, there is significant misalignment of the allowed momentum-energy manifolds corresponding to the two crystalline domains of graphene, which introduces a transport gap (usually 0.3 – 1.4 eV) that

depends exclusively on the periodicity^{2,3}. That is, class-II GB perfectly reflects low-energy carriers. In contrast, class-I GB is highly transparent with respect to charge carriers^{2,3}. Different from the strong dependence of charge carrier transport on GB type, the phonon transmission shows a weak dependence on GB type³. More importantly, both types of GBs greatly suppress the phonon transmission³. Therefore, the thermal conductivity change rate of graphene as a function of grain size is dramatically larger than the electrical conductivity change rate.

Considering the different change rates of electrical and thermal conductivity as a function of grain size, we deduce that nano-crystallization should be an efficient way to tune the electrical and thermal conductivities of polycrystalline graphene films for thermoelectric applications if graphene could be used in thermoelectric materials in the future. That is what we are trying to express in our manuscript. Recent theoretical calculations show that the polycrystalline graphene has a potential for thermoelectric materials, and the thermoelectric figure of merit of polycrystalline graphene can be enhanced by over three orders of magnitude above room temperature compared to crystalline graphene³. To be more accurate, we have changed the related statements about thermoelectric applications in the revised manuscript.

We have added the above discussions in our revised manuscript.

Reviewer #2 (Remarks to the Author):

After carefully evaluating the responses provided by the authors, I believe the changes in the revised manuscript have addressed all the major issues raised by the reviewers.

In my opinion the revised manuscript can be published in its present form.

Response: We thank the reviewer very much for the positive comments.

Reviewer #3 (Remarks to the Author):

The authors have satisfactorily addressed my concerns.

The only additional suggestion I have is the following. I'm not sure how to interpret the transport gap that is extracted. I assume it sets some energy scale for GB scattering. A more useful quantity would be GB resistivity, which the authors can probably determine through some simple modeling. I would like to see a comparison of this number against previous works (Huang, Yu, Tsen, etc.).

I recommend publication after this addition.

Response: We thank the reviewer very much for the constructive suggestions.

The transport gap refers to the energy that is needed to overcome for the charge carrier transmitting through the GB region. According to the reviewer's suggestion, we also fit the electronic transport data using the equation $R_s = R_s^G + \rho_{GB}/l_g^4$ (Figure R2), where R_s is the sheet resistance of the polycrystalline graphene films, R_s^G is the sheet resistance within the grain, ρ_{GB} is the GB resistivity, and l_g is the grain size. The fitting gives $R_s^G \approx 0.98 \text{ k}\Omega/\square$ and $\rho_{GB} \approx 0.33 \text{ k}\Omega\cdot\mu\text{m}$. It is worth noting that the GB resistivity extracted here is smaller than those reported previously, typically larger than $0.5 \text{ k}\Omega\cdot\mu\text{m}$ (Cummings et al⁴, $0.67 \text{ k}\Omega\cdot\mu\text{m}$; Yu et al⁵, $8 \text{ k}\Omega\cdot\mu\text{m}$; Tsen et al⁶, $1 \sim 40 \text{ k}\Omega\cdot\mu\text{m}$; Duong et al⁷, $1.4 \text{ k}\Omega\cdot\mu\text{m}$), further confirming the perfect stitching of neighboring grains in our graphene films. By using electrostatic force microscopy,

Huang et al⁸ tested the resistivity of individual GB of suspended graphene films. By assuming that the GB runs perpendicular to the line scan, they estimated an upper bound on the GB resistance of $60 \Omega \mu\text{m}/L$, where L is the length of the GB. Such smaller GB resistivity than ours and those reported by others⁴⁻⁷ is probably due to the use of suspended graphene films and the different measurement method.

We have added the above results and discussions in the revised manuscript.

Figure R2. Sheet resistance as a function of the inverse of grain size with a fit (red curve).

References:

1. Yazyev, O. V. & Chen, Y. P. Polycrystalline graphene and other two-dimensional materials. *Nat. Nanotechnol.* **9**, 755-767 (2014).
2. Yazyev, O. V. & Louie, S. G. Electronic transport in polycrystalline graphene. *Nat. Mater.* **9**, 806-809 (2010).

3. Lehmann, T., Ryndyk, D. A. & Cuniberti, G. Enhanced thermoelectric figure of merit in polycrystalline carbon nanostructures. *Phys. Rev. B* **92**, 035418 (2015).
4. Cummings, A. W. *et al.* Charge transport in polycrystalline graphene: Challenges and opportunities. *Adv. Mater.* **26**, 5079-5094 (2014).
5. Yu, Q. *et al.* Control and characterization of individual grains and grain boundaries in graphene grown by chemical vapour deposition. *Nat. Mater.* **10**, 443-449 (2011).
6. Tsen, A. W. *et al.* Tailoring electrical transport across grain boundaries in polycrystalline graphene. *Science* **336**, 1143-1146 (2012).
7. Duong, D. L. *et al.* Probing graphene grain boundaries with optical microscopy. *Nature* **490**, 235-239 (2012).
8. Huang, P. Y. *et al.* Grains and grain boundaries in single-layer graphene atomic patchwork quilts. *Nature* **469**, 389-392 (2011).

Reviewers' Comments:

Reviewer #1 (Remarks to the Author)

The issues have been resolved. I would like to recommend its publication.

Responses to the Reviewers' Comments

Reviewer #1 (Remarks to the Author):

The issues have been resolved. I would like to recommend its publication.

Response: We thank the reviewer very much for the positive comments.